# A Teacher-Guided Framework for Graph Representation Learning

## Abstract

We consider the problem of unsupervised representation learning for Graph Neural Networks (GNNs). Several state-of-the-art approaches to this problem are based on Contrastive Learning (CL) principles that generate transferable representations. Their objective function can be posed as a supervised discriminative task using 'hard labels,' as they consider each pair of graphs as either 'equally positive' or 'equally negative'. However, it has been observed that using 'soft labels' in a Bayesian way can reduce the variance of the risk for discriminative tasks in supervised settings. Motivated by this, we propose a CL framework for GNNs, called *Teacher-guided Graph Contrastive Learning (TGCL)*, that incorporates 'soft labels' to facilitate a more regularized discrimination. In particular, we propose a teacher-student framework where the student network learns the representation by distilling the representations produced by the teacher network trained using unlabelled graphs. Our proposed approach can be adapted to any existing CL methods and empirically improves the performance across diverse downstream tasks.

## 1 Introduction

Graphs are versatile data structures representing relationships between entities in various real-world applications, *e.g.*, social networks (Ohtsuki et al., 2006; Fan et al., 2019), bio-informatics (Muzio et al., 2021), and knowledge graphs (Wang et al., 2014; Baek et al., 2020). Acquiring labeled information for these applications can be a costly and time-consuming process. Towards this, *self-supervised Learning (SSL)* has emerged as an important research area that leverages the inherent structure or content of the data to learn informative representations without relying on explicit labels (Hu* et al., 2020; Hwang et al., 2020; Grover & Leskovec, 2016).

Existing SSL methods for graphs can be broadly categorized as: **(a)** *local similarity-based predictive learning* and **(b)** *global similarity-based contrastive learning*. *Predictive learning*-based methods (Hu* et al., 2020; Kim & Oh, 2021; Rong et al., 2020) produces artificial labels by capturing specific local contextual information of neighborhood sub-graphical features to produce the representations. However, it restricts them only to capture the local graph semantics. Alternatively, *contrastive learning (CL)*-based models for graphs aim to maximize the agreements between instances perturbed by *semantic-invariant augmentations* (positive views) while repelling the others (negative views) to capture global semantics. CL-based SSL models are extremely popular in the computer-vision community as one can easily generate such semantic-invariant perturbations using simple techniques *e.g.,* rotation, flipping, and color jittering (Chen et al., 2020a; Grill et al., 2020). Several graph contrastive learning methods are also proposed where the positive pairs are produced using transformations *e.g.,* edge perturbation, attribute masking, and subgraph sampling. However, unlike continuous domains (e.g., images), even "minor" modifications in the graph structures, such as removing one edge or node, can significantly change the properties of graphs due to their discrete nature (see Figure 1a and 1b). Recently, *discrepancy-based Self-supervised Learning (D-SLA)* (Kim et al., 2022) introduces edit distance-based discrepancy measures to address these issues. However, computing the edit distance between two arbitrary graphs is NP-hard (Sanfeliu & Fu, 1983; Zeng et al., 2009). Further, it can only provide high-level structural information without capturing any semantic differences (Figure 1a and Figure 1b). Towards this, we aim to develop a graph representation learning framework that incorporates more *semantically-rich* discriminative features to regularize the learning.

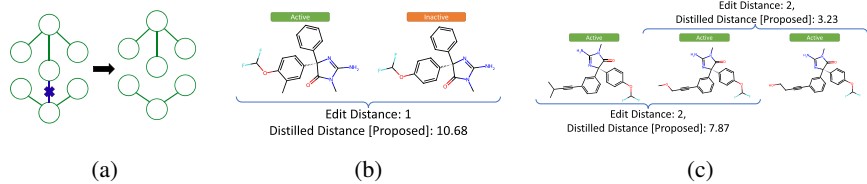

(a)                            (b)                            (c)

Figure 1: Illustrating the shortcomings of existing CL methods: **(a)** Even a "minor change" of removal of one edge significantly changes the graph to two disconnected components that cannot be captured even using edit-distance-based discrepancy (Kim et al., 2022). **(b & c)** Next, we see a more specific example of correlated-structured molecules that either actively bind to a set of human $\beta$-secretase inhibitors or inactive (Wu et al., 2018). **(b)** Molecules having dissimilar properties can have smaller edit distances while **(c)** molecules from the same class can have larger edit distances. In other words, edit distance remains ineffective in capturing chemical semantics. To this end, we propose a *distilled distance* form a pre-trained teacher network to incorporate a "soft" perception of semantic distances for arbitrary graphs to produce better representations for a student network.

## 1.1 MOTIVATION & CONTRIBUTIONS

The existing CL methods can be viewed under one umbrella since all of these techniques aim to learn representations by contrasting different views of the input graphs. In principle, these loss functions can be considered classification objectives by creating pseudo-labels among different views of input graphs Oord et al. (2018); Gutmann & Hyvärinen (2010). On the other hand, in the supervised learning literature, it has been observed that incorporating *'soft labels'* in the form of *Knowledge Distillation (KD)* leads to better generalization Menon et al. (2021); Hinton et al. (2015). Given these prior results, we ask the following question in this work:

*Whether introducing 'soft labels' for CL methods can produce better graph representations?*

The fundamental idea of KD is to use softened labels via a teacher network while minimizing the supervised risk of a student network by reducing the divergence between their logits Hinton et al. (2015). Prior works have shown that *Bayes-distilled risk has lower variance* compared to naive undistilled counterpart, which leads to better generalization Menon et al. (2021). Motivated by these results, we propose a novel *Teacher-guided Graph Contrastive Learning (TGCL)* framework. We design a *distilled perception distance* (or *distilled distance*) between two arbitrary input graphs using their deep features obtained from a pre-trained "teacher" to define a softer notion of positive/negative pairs. We train the student network by incorporating such 'soft labels' for each pair of graphs. We argue that by introducing distilled distance, we can introduce the regularized semantic difference between two arbitrary graphs, addressing the shortcomings of the existing CL frameworks for graphs. For example, Figure 1c demonstrates that our distilled distance obtained from the "teacher" can significantly differ among molecular graphs with correlated structures, towards capturing the chemical semantic differences for graphs. Figure 1b shows that the distilled distance captures the chemical semantic difference of molecules with different chemical properties, however, with a minor structural difference. The contributions of this work are summarized below:

1. We propose a novel *Teacher-guided Graph Contrastive Learning (TGCL)* framework for incorporating the concept of knowledge distillation to learn graph representations.

2. The concept of "soft-labeled" pairs of graphs can be applied to any contrastive learning framework. We propose a *distilled perception distance* and present two TGCL frameworks by modifying the well-known NT-Xent loss and D-SLA method to incorporate smooth perception from a teacher network for training the student network.

3. We conduct extensive experiments on graph classification for molecular datasets and link prediction on social network datasets where our proposed framework consistently outperforms the existing methods.

## 2 RELATED WORK

### 2.1 REPRESENTATION LEARNING ON GRAPHS

***Classical Approaches:*** A straightforward graph representation approach considers a *'bag of nodes'*, which falls short in capturing their overall semantics (Hamilton, 2020). Weisfeiler-Lehman kernel (Shervashidze et al., 2011) improves upon this by utilizing the iterative neighborhood aggregation strategy. One may also count the occurrence of small subgraph structures, called graphlets. However, it is a combinatorially challenging problem, and approximate algorithms are required (Ahmed et al., 2015; Hočevar & Demšar, 2014). A few other approaches enumerate different kinds of paths in graphs (Kashima et al., 2003; Borgwardt & Kriegel, 2005).

***Shallow Algorithms:*** DeepWalk Perozzi et al. (2014) and LINE Tang et al. (2015) are random walk-based approaches using depth-first search (DFS) and breadth-first search (BFS) algorithms, respectively. "node2vec" Grover & Leskovec (2016) combines both BFS and DFS to learn node embeddings that maximize the likelihood of preserving node neighborhoods.

***Predictive Self-supervised Graph Representation Learning:*** Here, the model aims to predict specific properties or relationships of graphs, such as predicting the attributes of masked nodes/edges (Hu* et al., 2020), predicting the presence of an edge (Hwang et al., 2020), or predicting contextual properties and presence of motifs (Hu et al., 2020; Rong et al., 2020). These predictive tasks serve as self-supervised learning objectives, as they do not require explicit supervised labels. Instead, they rely on the local sub-structure of the graph for producing labels.

***Contrastive Self-supervised Graph Representation Learning:*** Deep Graph Infomax (DGI) (Veličković et al., 2019) maximizes the mutual information between graph representation and patch representation. InfoGraph (Sun et al., 2020) maximizes the mutual information between the graph-level representation and the representations of substructures of different scales, such as nodes, edges, and triangles. Several other works (You et al., 2020; 2021; Zhu et al., 2021; Yin et al., 2022; Wang et al., 2022) generate a perturbed view of the original graph through attribute masking, edge perturbation, and subgraph sampling and employ contrastive learning framework for better representation learning. Recent works Suresh et al. (2021); Yang et al. (2021) also explore adversarial augmentation strategies to improve the contrastive frameworks' representations further.

### 2.2 KNOWLEDGE DISTILLATION (KD)

KD (Hinton et al., 2015) is a popular technique to transfer knowledge from a large, complex model (*a.k.a.* the teacher) to a smaller, efficient model (*a.k.a.* the student) while performing similarly to the teacher. Several ongoing works also focus on improving the student's performance on a wide range of applications (Heo et al., 2019; Furlanello et al., 2018; Lopes et al., 2017; Li et al., 2021; Lee et al., 2018; Bhat et al., 2021). KD allows the student to learn from both the raw data and distilled knowledge of the teacher, improving their generalized performance (Menon et al., 2021). A comprehensive review of KD can be found in (Wang & Yoon, 2021).

***Self-supervised learning with KD*** is previously explored for computer vision, typically to improve the performance of smaller models (Abbasi Koohpayegani et al., 2020; Chen et al., 2020b). Many of these approaches combined KD with CL methods (Fang et al., 2021; Gao et al., 2022). SimCLR-v2 (Chen et al., 2020b) applied a larger teacher model, first trained using contrastive loss followed by supervised fine-tuning to distill a smaller model via self-supervised learning. Xu et al. (2020) incorporates auxiliary contrastive loss to obtain richer knowledge from the teacher network. Other approaches are proposed to transfer the final embeddings of a self-supervised pre-trained teacher (Navaneet et al., 2022; Song et al., 2023).

***Limitations & Challenges:*** (1) Most of these methods applied supervised labels to finetune the teacher network, restricting their applicability. (2) Further, these methods may not be applicable to graph inputs due to their discrete structure where even minor perturbations can significantly change their semantics. Towards this, our proposed TGCL first aims to obtain the teacher's distilled perception to calculate the semantic difference for any pairs, followed by formulating soft self-supervised losses to train the student.

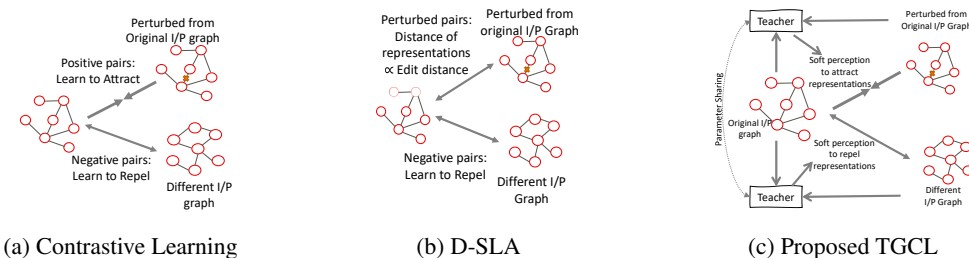

(a) Contrastive Learning      (b) D-SLA      (c) Proposed TGCL

Figure 2: Comparing our proposed teacher-guided contrastive learning (TGCL) framework with the existing contrastive learning (You et al., 2020; Xu et al., 2021), and D-SLA (Kim et al., 2022). From the classification point of view, the standard CL methods consider the similarity between the anchor and the perturbed graphs as "hard" positive pairs while the other graphs as "hard" negative pairs. D-SLA introduces "hard" discrepancies using edit distance between the anchor and the perturbed graphs, while the other graphs as "hard" negative pairs. Our proposed TGCL introduces a novel distilled perception distance for smooth discrimination between any arbitrary graphs.

## 3    PROPOSED METHOD

### 3.1    PRELIMINARIES

***Graph Neural Network (GNN).*** Let $G = (V, E, X_V, X_E)$ be an undirected graph in the space of graphs $\mathcal{G}$, where $V, E, X_V, X_E$ denote the set of nodes, edges, node attributes, and edge attributes respectively. GNN encodes a graph $G \in \mathcal{G}$ to a $d$-dimensional embedding vector: $f : \mathcal{G} \rightarrow \mathbb{R}^d$. $f$ is often composed by stacking multiple message-passing layers. Let $h_v^{(l)}$ denote the representation of a node $v \in V$ having a neighborhood $N_v$ in the $l^{th}$ layer. $h_{vu}^{(l-1)}$ represents the attributes of edge $(v, u) \in E$ in the $(l-1)^{th}$ layer. Then, $h_v^{(l)}$ can be expressed as follows:

$$h_v^{(l)} = \phi_U^{(l-1)}\left(h_v^{(l-1)}, \bigoplus_{u \in N_v} \psi_M^{(l-1)}\big(h_v^{(l-1)}, h_u^{(l-1)}, h_{vu}^{(l-1)}\big)\right), \tag{1}$$

where $\phi_U^{(l-1)}, \psi_M^{(l-1)}$ are the update and the message function of $(l-1)^{th}$ layer respectively. $\oplus$ is a permutation invariant aggregation operator.

***Global Representations for Graphs using Contrastive Learning.*** Contrastive learning (CL) aims to learn meaningful representations by attracting the positive pairs (*i.e.,* similar instances, such as two different perturbations of the same graph) while repelling negative pairs (*i.e.,* dissimilar instances, such as two different input graphs) in an unsupervised manner, as shown in Figure 2(b). Formally, let $G_0$ denote the original graph, $G_p$ denote its perturbed version (*i.e.,* a positive pair), and $G_n$ is a different input graph (*i.e.,* negative sample). Then, the CL objective can be defined as follows:

$$\mathcal{L}_{CL} = -\log \frac{\text{sim}\big(f(G_0), f(G_p)\big)}{\sum_{G_n} \text{sim}(f(G_0), f(G_n))}, \tag{2}$$

where $f$ is a GNN and $\text{sim}(\cdot, \cdot)$ is a similarity measure for embeddings.

Minimization of Equation 2 brings positive pairs closer and pushes negative pairs further apart in the embedding space. However, unlike image augmentation schemes (*e.g.,* scaling, rotation, color jitter), graph augmentation schemes (*e.g.,* node/edge perturbations, subgraph sampling) may fail to preserve the graph semantics. For example, Figure 2b illustrates that removing one edge leads to two disconnected graphs, significantly changing the original semantics. Recently, D-SLA incorporates edit distances between graphs to train their model, partially addressing this issue (Kim et al., 2022).

### 3.2    PROPOSED TEACHER-GUIDED CONTRASTIVE LEARNING (TGCL) FOR GRAPHS

The fundamental motivation of our proposed TGCL framework is based on the following results. More formal statements can be found in the Appendix B.

- *Noise contrastive Estimation, which is equivalent to solving a binary classification problem between the samples of data and noise (Gutmann & Hyvärinen, 2010)[Proposition B.1].*

- *In supervised learning, 'soft labels' in the form of Knowledge Distillation (KD) leads to better generalization by reducing the variance of Bayes-distilled risk (Menon et al., 2021)* [Proposition B.2 and B.3].

The first result indicates that the existing CL methods can be viewed as a supervised classification loss where the network is trained by producing artificially generated "hard pseudo-labels" Gutmann & Hyvärinen (2010); Oord et al. (2018). For example, $\mathcal{L}_{cl}$ (in Eq. 2) can be viewed as labeling the similarity between positive pairs, $\text{sim}(f(G_0), f(G_p))$ as a positive class, and negative pairs are considered as the negative class. Similarly, we can analyze the loss components for D-SLA (Kim et al., 2022): Their graph discrimination loss considers the original and perturbed graphs as two different classes. Their edit-distance based-loss uses the edit distance between the anchor and the perturbed graph as a "hard margin" to learn the representations. Finally, their margin loss acts similar to $\mathcal{L}_{cl}$ (Eq. 2) where the similarity between the anchor and the perturbed graph is labeled as 1, and the similarity between two arbitrary graphs is labeled as 0.

The second result demonstrates that we can achieve better generalization performance by 'softening' the labels for an existing CL method as the Bayes-distilled risk has lower variance compared to the naive un-distilled counterpart (Menon et al., 2021). To this end, we introduce a softer notion of distance by proposing a distilled perception distance, $\mathcal{D}_{dp}$ for any two arbitrary graphs by comparing their deep features from a teacher network, pre-trained using 'hard' pseudo-labels.

***Distilled Perceptual Distance.*** Let $G_a$ and $G_b$ be two arbitrary graphs. Consider a representation learning model with $L$ message passing layers as the teacher. At each layer, $l$, we obtain the node-embedding $\{h_v^{(l)}\}_{v \in V}$ for a graph $G$ and apply a pooling operation (*e.g.*, max-pool, avg-pool) to obtain a fixed-length vector, denoted as $h_G^{(l)}$. We extract such fixed-length features from each layer and concatenate them, *i.e.*, $h_{G_a} = [\{h_{G_a}^{(l)}\}_l]$ and $h_{G_b} = [\{h_{G_b}^{(l)}\}_l]$ for $G_a$ and $G_b$ respectively. The *distilled perception distance (or distilled distance)* $\mathcal{D}_{dp}$ is then defined as the $L_2$ distance between these concatenated features, as follows:

$$\mathcal{D}_{dp}(G_a, G_b) = ||h_{G_a} - h_{G_b}||_2 \qquad (3)$$

Notably, our proposed *distilled distance* is similar to the well-known "perceptual distance" from computer vision literature. It compares the high-level rich latent activations extracted from a pre-trained convolutional neural network (CNN) (*e.g.*, VGG (Simonyan & Zisserman, 2014) or ResNet (He et al., 2016)) to incorporate semantic differences between two samples (Johnson et al., 2016).

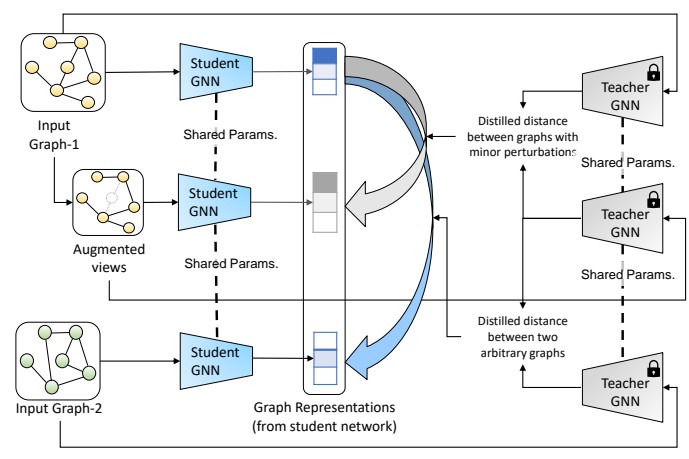

Figure 3: Block diagram of our proposed TGCL framework. We obtain the representations from a pre-trained teacher model and compute the distilled distance for each pair of inputs. These pairwise distances are employed to "soften" the loss functions to train the student.

### 3.3 PROPOSED LOSS FUNCTIONS

The concept of teacher-guided loss with "softer" positive/negative pairs to train the student network can be introduced to any contrastive learning framework for graphs. To showcase the flexibility of our proposed TGCL framework, we present two versions of our TGCL framework using normalized temperature-scaled cross-entropy (NT-Xent) (Chen et al., 2020a; You et al., 2020) loss and D-SLA (Kim et al., 2022): **(I)** *TGCL-NTXent* & **(II)** *TGCL-DSLA*.

### 3.3.1 TGCL-NTXENT: TGCL FRAMEWORK USING NT-XENT LOSS

We modify the NT-Xent loss for our TGCL framework as follows:

$$\mathcal{L}_{TGCL-NTXent} = \sum_{G_{p_i}} -\log \frac{\exp(\mathcal{D}_{dp}(G_0, G_{p_i}) \cdot f_s(G_0) \cdot f_s(G_{p_i})/\tau)}{\sum_{G_{n_j}} \exp(\mathcal{D}_{dp}(G_0, G_{n_j}) \cdot f_s(G_0) \cdot f_s(G_{n_j})/\tau)} \tag{4}$$

where, $f_s$ is the student network and $f_s(\cdot)$ is the representations obtained from $f_s$. $G_0$ is the anchor sample, $G_{p_i}$ is the $i^{th}$ perturbed sample. $G_{n_j}$ is $j^{th}$ negative sample for the anchor, $G_0$.

Here, we incorporate distilled distance, $\mathcal{D}_{dp}(G_0, G_p)$, in both numerator and denominator. The intuition is to produce larger similarly for positive pairs (i.e., $f_s(G_0) \cdot f_s(G_p)$) when the teacher's perception of distilled distance is small.

### 3.3.2 TGCL-DSLA: TGCL FRAMEWORK USING D-SLA

Next, we demonstrate how to introduce distilled perception distance to incorporate the concept of teacher-guided "soft pairs" for D-SLA.

**(a) Teacher-guided Soft Discrimination:** We first discriminate the perturbed graphs from the original anchor by introducing $\mathcal{L}_{T-Soft}$: It consists of two terms: the first one is a KD-based loss, $\mathcal{L}_{KD}$, while the second component is a weighted graph discrimination loss ($\mathcal{L}_{wGD}$). We first obtain the distilled distances: $[\mathcal{D}_{dp}(G_0, G_0), \{\mathcal{D}_{dp}(G_0, G_{p_i})\}_i]$ between the anchor, $G_0$, with itself and the $i^{th}$ perturbed variations, $G_{p_i}$. We obtain the similarities by taking reciprocals of the normalized distilled distance, followed by clipping to ensure numerical stability:

$$s_0 = \text{clip}(\mathcal{D}_{dp}(G_0, G_0)^{-1}) \quad \text{and} \quad s_i = \text{clip}(\mathcal{D}_{dp}(G_0, G_{p_i})^{-1}) \ \forall i \tag{5}$$

Next, we compute a probability distribution (soft labels) using the softmax-activation with temperature, $\tau$ i.e., $\text{softmax}(s_0, s_1, \cdots; T = \tau)$. Similarly, we obtain a score for each graph and compute a probability distribution using temperature-scaled softmax: $\text{softmax}(\Psi \circ f_s(G_{p_i}); T = \tau)$. Now, we obtain the distillation loss, $\mathcal{L}_{KD}$ by minimizing the entropy between these probability distributions:

$$\mathcal{L}_{KD} := \tau^2 \mathcal{H}\Big(\text{softmax}(s_0, s_1, \cdots; \tau), \text{softmax}(\Psi \circ f_s(G_{p_i}); \tau)\Big) \tag{6}$$

where, $\mathcal{H}(y, \hat{y}) = \sum_y -y \log \hat{y}$ is the cross-entropy function. $\Psi \circ f_s$ is the composition of the score function, $\Psi$, and the student network, $f_s$. Therefore, this loss incorporates the teacher's smoothened perception in the score functions to learn the student's representations.

The second term, $\mathcal{L}_{wGD}$, is a set of *binary cross-entropy* functions with $G_0$ is labeled as 1 and $G_{p_i}$s' are labeled as 0 with the associated soft-weights as $w_i$:

$$\mathcal{L}_{wGD} = \Big[\mathcal{H}(1, \sigma(\Psi \circ f_s(G_0))) + \sum_i w_i \mathcal{H}(0, \sigma(\Psi \circ f_s(G_{p_i})))\Big], \tag{7}$$

where, $w_i = \frac{\mathcal{D}_{dp}(G_0, G_{p_i})}{\sum_i \mathcal{D}_{dp}(G_0, G_{p_i})}$. Therefore, $\mathcal{L}_{wGD}$ incorporates the teacher's soft label via $w_i$. Now, $\mathcal{L}_{T-soft}$ combines both aforementioned loss components with a hyper-parameter $\alpha$.

$$\mathcal{L}_{T-soft} = \alpha \mathcal{L}_{KD} + (1 - \alpha)\mathcal{L}_{wGD} \tag{8}$$

**(b) Teacher-guided Perception Loss:** Next, we introduce a perception loss, $\mathcal{L}_{T-percept}$. It ensures that the embedding-level difference between original and perturbed graphs is proportional to a teacher's perspective of their corresponding distilled distances.

$$\mathcal{L}_{T-percept} = \sum_{i,j} \left( \frac{\text{dist}(f_s(G_{p_i}), f_s(G_0))}{\mathcal{D}_{dp}(G_{p_i}, G_0)} - \frac{\text{dist}(f_s(G_{p_j}), f_s(G_0))}{\mathcal{D}_{dp}(G_{p_j}, G_0)} \right)^2 \tag{9}$$

where, $\text{dist}(f_s(G_a), f_s(G_b))$ denotes the $L_2$ distance of graph $G_a$ $G_b$ in their representation space.

**(c) Teacher-guided Margin Loss for Negative Graphs:** Our third component, $\mathcal{L}_{T-Margin}$ is a modified margin loss where the distilled distance acts as a regularizer, controlling the margin among

Table 1: Performance (mean ± std) comparison on graph classification task.

| | Methods | BBBP | ClinTox | MUV | HIV | BACE | SIDER | Tox21 | ToxCast | Avg |
|---|---|---|---|---|---|---|---|---|---|---|
| | No Pretrain | 65.8 ± 4.5 | 58.0 ± 4.4 | 71.8 ± 2.5 | 75.3 ± 1.9 | 70.1 ± 5.4 | 57.3 ± 1.6 | 74.0 ± 0.8 | 63.4 ± 0.6 | 66.96 |
| Predictive | Edgepred Hamilton et al. (2017) | 67.3 ± 2.4 | 64.1 ± 3.7 | 74.1 ± 2.1 | 76.3 ± 1.0 | 79.9 ± 0.9 | 60.4 ± 0.7 | 76.0 ± 0.6 | 64.1 ± 0.6 | 70.28 |
| | AttrMasking Hu* et al. (2020) | 64.3 ± 2.8 | 71.8 ± 4.1 | 74.7 ± 1.4 | 77.2 ± 1.1 | 79.3 ± 1.6 | 61.0 ± 0.7 | 76.7 ± 0.4 | 64.2 ± 0.5 | 71.15 |
| | ContextPred Hu* et al. (2020) | 68.0 ± 2.0 | 65.9 ± 3.8 | 75.8 ± 1.7 | 77.3 ± 1.0 | 79.6 ± 1.2 | 60.9 ± 0.6 | 75.7 ± 0.7 | 63.9 ± 0.6 | 70.89 |
| Contrastive | Infomax Veličković et al. (2019) | 68.8 ± 0.8 | 69.9 ± 3.0 | 75.3 ± 2.5 | 76.0 ± 0.7 | 75.9 ± 1.6 | 58.4 ± 0.8 | 75.3 ± 0.5 | 62.7 ± 0.4 | 70.29 |
| | GraphCL (You et al., 2020) | 69.7 ± 0.7 | 76.0 ± 2.7 | 69.8 ± 2.7 | 78.5 ± 1.2 | 75.4 ± 1.4 | 60.5 ± 0.9 | 73.9 ± 0.7 | 62.4 ± 0.6 | 70.78 |
| | JOAO (You et al., 2021) | 70.2 ± 1.0 | 81.3 ± 2.5 | 71.7 ± 1.4 | 76.7 ± 1.2 | 77.3 ± 0.5 | 60.0 ± 0.8 | 75.0 ± 0.3 | 62.9 ± 0.5 | 71.89 |
| | JOAOv2 (You et al., 2021) | 71.4 ± 0.9 | 81.0 ± 1.6 | 73.7 ± 1.0 | 77.7 ± 1.2 | 75.5 ± 1.3 | 60.5 ± 0.7 | 74.3 ± 0.6 | 63.2 ± 0.5 | 72.16 |
| | GraphLoG (Xu et al., 2021) | 72.5 ± 0.8 | 76.7 ± 3.3 | 76.0 ± 1.1 | 77.8 ± 0.8 | 83.5 ± 1.2 | 61.2 ± 1.1 | 75.7 ± 0.5 | 63.5 ± 0.7 | 73.36 |
| | BGRL (Thakoor et al., 2022) | 66.7 ± 1.7 | 64.7 ± 6.5 | 69.4 ± 2.7 | 75.5 ± 1.9 | 71.3 ± 5.5 | 60.4 ± 1.4 | 74.8 ± 0.7 | 63.2 ± 0.8 | 68.25 |
| | SimGCL (Yu et al., 2022) | 67.4 ± 1.2 | 55.7 ± 4.7 | 71.2 ± 1.8 | 75.0 ± 0.9 | 74.1 ± 2.7 | 57.4 ± 1.7 | 74.4 ± 0.5 | 62.3 ± 0.4 | 67.19 |
| | SimGRACE (Xia et al., 2022) | 71.3 ± 0.9 | 64.2 ± 4.5 | 71.2 ± 3.4 | 74.5 ± 1.1 | 73.8 ± 1.4 | 60.59 ± 0.9 | 74.2 ± 0.6 | 63.4 ± 0.5 | 69.13 |
| | D-SLA (Kim et al., 2022) | 72.6 ± 0.8 | 80.2 ± 1.5 | 76.6 ± 0.9 | 78.6 ± 0.4 | 83.8 ± 1.0 | 60.2 ± 1.1 | 76.8 ± 0.5 | 64.2 ± 0.5 | 74.13 |
| Ours | TGCL-NTXent (w/ GraphLoG) | **74.9 ± 0.9** | **85.3 ± 2.2** | 78.9 ± 1.0 | **79.1 ± 0.5** | 83.7 ± 1.4 | 63.6 ± 0.6 | 76.7 ± 0.4 | 64.1 ± 0.4 | **75.79** |
| | TGCL-NTXent (w/ D-SLA) | 74.0 ± 0.4 | 82.8 ± 2.2 | 77.0 ± 0.9 | 77.9 ± 0.3 | 84.3 ± 1.0 | **64.2 ± 0.3** | 76.6 ± 0.1 | 64.7 ± 0.4 | 75.19 |
| Ours | TGCL-DSLA (w/ GraphLoG) | 74.8 ± 0.3 | 80.6 ± 0.5 | 77.4 ± 0.1 | 78.6 ± 0.2 | 83.0 ± 1.1 | 61.4 ± 0.4 | 76.1 ± 0.1 | 64.0 ± 0.3 | 74.49 |
| | TGCL-DSLA (w/ D-SLA) | 73.5 ± 0.9 | **84.9 ± 1.3** | **79.4 ± 0.9** | 78.8 ± 0.5 | **85.2 ± 0.4** | 61.2 ± 1.0 | **76.9 ± 0.1** | **64.9 ± 0.2** | 75.60 |

the anchor, $G_0$, its perturbed variations, $G_{p_i}$ and the negative graphs, $G_{n_j}$, as follows:

$$\beta_{ij} = \max\big(\beta,\ \mathcal{D}_{dp}(G_0, G_{n_j}) - \mathcal{D}_{dp}(G_0, G_{p_i})\big) \tag{10}$$

$$\mathcal{L}_{T-Margin} = \sum_{i,j} \max\left(0, \beta_{ij} + \text{dist}\big(f_s(G_{p_i}), f_s(G_o)\big) - \text{dist}\big(f_s(G_{n_j}), f_s(G_o)\big)\right) \tag{11}$$

***Overall Loss:*** We obtain the overall loss by combining all three components as follows:

$$\mathcal{L}_{TGCL-DSLA} = \mathcal{L}_{T-soft} + \lambda_1 \mathcal{L}_{T-percept} + \lambda_2 \mathcal{L}_{T-Margin} \tag{12}$$

Where $\lambda_1$ and $\lambda_2$ are the hyper-parameters for training the student model, as used in D-SLA.

## 4 EXPERIMENTAL RESULTS

An effective representation of a graph should capture both the global structure and the local semantics. Therefore, to gauge the efficacy of the proposed method in learning informative representations, we conduct two sets of experiments — (i) Graph Classification and (ii) Link prediction. We provide additional details and ablation studies in our Appendix. Our anonymized code is available [here].

### 4.1 GRAPH CLASSIFICATION

***Datasets.*** Following prior works (You et al., 2021; Xu et al., 2021; Kim et al., 2022), we utilize ZINC15 (Sterling & Irwin, 2015) to train the representation learning models. Next, we finetune the models on eight different molecular benchmarks from MoleculeNet (Wu et al., 2018). We divide the datasets based on the constituting molecules' scaffold (molecular substructure) and evaluate models' generalization ability on out-of-distribution test data samples (Wu et al., 2018).

***State-of-the-art Baselines.*** We compare the proposed method's performance against twelve existing methods: EdgePred (Hamilton et al., 2017), AttrMasking (Hu* et al., 2020), ContextPred (Hu* et al., 2020), Infomax (Veličković et al., 2019), GraphCL (You et al., 2020), JOAO (You et al., 2021), JOAOv2 (You et al., 2021), GraphLoG (Xu et al., 2021), BGRL (Thakoor et al., 2022), SimGCL (Yu et al., 2022), SimGRACE (Xia et al., 2022) and D-SLA (Kim et al., 2022).

***Evaluation Metric.*** We compare the *Area Under Receiver Operating Characteristic curve (AU-ROC)* for benchmarking (Davis & Goadrich, 2006). AUROC quantifies the overall discriminative power of the classifier across all possible classification thresholds. AUROC value ranges between 0 and 1, with higher values indicating better discrimination ability of the model.

***Results.*** In Table 1, we can see that the *"no pretraining"* model achieves the least performance. While predictive pretraining improves upon the no pertaining model, their performance remains worse than the CL models. This is because predictive methods primarily focus on the local structure, while molecule property classification requires global structure. In contrast, CL methods focus on the global structure by contrasting between original and perturbed graphs to achieve better performance. While a few augmentation-free CL (Yu et al., 2022; Xia et al., 2022) methods are proposed,

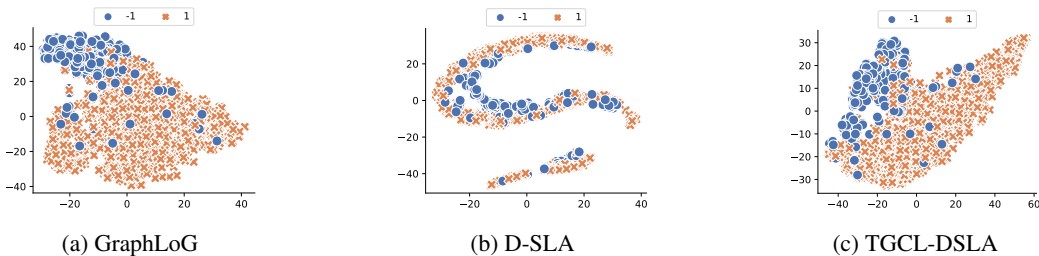

|  (a) GraphLoG | (b) D-SLA | (c) TGCL-DSLA |

Figure 4: t-SNE visualization of learned representations using for BBBP dataset.

their performance remains significantly lower than the state-of-the-art. GraphLoG achieves near state-of-the-art performance by exploring both global semantics and local substructures. However, D-SLA achieves state-of-the-art performance by exploring the local discrete properties of graphs.

For our proposed framework, we report the results by using GraphLoG and D-SLA as teacher modules for both TGCL-NTXent and TGCL-DSLA models, demonstrating the generalizability of our framework. As we can see, the proposed method achieves a consistent performance boost irrespective of the training methodology of the teacher module. Furthermore, we observe that TGCL-NTXent (w/ GraphLoG) achieves the best performance. Also, TGCL-DSLA (w/ D-SLA) achieves comparable performance as TGCL-NTXent (w/ GraphLoG). In particular, we do not observe an additional advantage of using more graph-specific D-SLA-based loss functions while learning global graph-level representations for molecular property prediction tasks.

In Figure 4, we visualize the learned latent space of GraphLoG, D-SLA, and our TGCL-DSLA (w/ DSLA) utilizing t-SNE (Van der Maaten & Hinton, 2008). We observe that both GraphLog and TGCL-DSLA segregate the positive and negative samples more successfully compared to DSLA.

## 4.2 LINK PREDICTION

***Datasets.*** We consider COLLAB, IMDB-Binary, and IMDB-Multi from TU Benchmarks (Morris et al., 2020). We use the same data splits as in (Kim et al., 2022).

***State-of-the-art Baselines.*** We compare the proposed method against various predictive methods (AttrMasking (Hu* et al., 2020), ContextPred (Hu* et al., 2020)), contrastive frameworks (Infomax (Veličković et al., 2019), GraphCL (You et al., 2020), JOAO (You et al., 2021), GraphLoG (Xu et al., 2021), BGRL (Thakoor et al., 2022)), SimGCL (Yu et al., 2022), SimGRACE (Xia et al., 2022) and discriminative learning algorithm (D-SLA (Kim et al., 2022)).

***Evaluation Metric.*** We compare average precision (as in (Kim et al., 2022)). It ranges from 0 to 1, with a higher value indicating better performance.

***Results.*** Table 2 demonstrates that, unlike graph classification tasks, local context plays a crucial role in link prediction. Therefore, the predictive models typically outperformed most of the CL methods. Among the existing CL methods, GraphLog performs similarly to ContextPred as it focuses on both local and global structures. D-SLA achieves better performance by capturing local structures using edit-distance-based discriminations that standard CL models fail to distinguish.

Table 2: Performance (mean ± std) comparison on link prediction task.

| Methods | COLLAB | IMDB-Binary | IMDB-Multi | Avg. |
|---|---|---|---|---|
| No Pretrain | 80.01 ± 1.14 | 68.72 ± 2.58 | 64.93 ± 1.92 | 71.22 |
| AttrMasking (Hu* et al., 2020) | 81.43 ± 0.80 | 70.62 ± 3.68 | 63.37 ± 2.15 | 71.81 |
| ContextPred (Hu* et al., 2020) | 83.96 ± 0.75 | 70.47 ± 2.24 | 66.09 ± 2.74 | 73.51 |
| Infomax (Veličković et al., 2019) | 80.83 ± 0.62 | 67.25 ± 1.87 | 64.98 ± 2.47 | 71.02 |
| GraphCL (You et al., 2020) | 76.04 ± 1.04 | 63.71 ± 2.98 | 62.40 ± 3.04 | 67.38 |
| JOAO (You et al., 2021) | 76.57 ± 1.54 | 65.37 ± 3.23 | 62.76 ± 1.52 | 68.23 |
| GraphLoG (Xu et al., 2021) | 82.95 ± 0.98 | 69.71 ± 3.18 | 64.88 ± 1.87 | 72.51 |
| BGRL (Thakoor et al., 2022) | 76.79 ± 1.13 | 67.97 ± 4.14 | 63.71 ± 2.09 | 69.49 |
| SimGCL (Yu et al., 2022) | 77.46 ± 0.86 | 64.91 ± 2.60 | 63.78 ± 2.28 | 68.72 |
| SimGRACE (Xia et al., 2022) | 74.51 ± 1.54 | 64.49 ± 2.79 | 62.81 ± 2.32 | 67.27 |
| D-SLA (Kim et al., 2022) | 86.21 ± 0.38 | 78.54 ± 2.79 | 69.45 ± 2.29 | 78.07 |
| TGCL-NTXent (w/ GraphLoG) | 87.23 ± 0.14 | 75.09 ± 1.88 | 67.11 ± 3.73 | 76.48 |
| TGCL-NTXent (w/ D-SLA) | 87.51 ± 1.24 | 77.95 ± 3.89 | 67.88 ± 2.20 | 77.78 |
| TGCL-DSLA (w/ GraphLoG) | **91.09 ± 0.33** | **83.15 ± 0.89** | **74.11 ± 1.44** | **82.78** |
| TGCL-DSLA (w/ D-SLA) | 87.51 ± 0.59 | 80.03 ± 4.13 | 70.97 ± 2.42 | 79.50 |

In comparison, our proposed distilled distance from the teacher network incorporates a regularized notion of both local and global semantics. The local semantics are encapsulated from the latent features of the initial layers, while the global semantics are contained within the high-level global features. Therefore, we can surpass existing local and global representation learning-based models by visible margins for all three datasets. Interestingly, our TGCL-DSLA (w/GraphLog) performs

Table 3: Impact of the capacity of the student TGCL-DSLA models (mean $\pm$ std) for graph classification. "Full-capacity" denotes the 5-layered model (i.e., the same capacity as the teacher).

|  | BBBP | ClinTox | MUV | HIV | BACE | SIDER | Tox21 | ToxCast | Avg |
|---|---|---|---|---|---|---|---|---|---|
| D-SLA Kim et al. (2022) | $72.6 \pm 0.8$ | $80.2 \pm 1.5$ | $76.6 \pm 0.9$ | $78.6 \pm 0.4$ | $83.8 \pm 1.0$ | $60.2 \pm 1.1$ | $76.8 \pm 0.5$ | $64.2 \pm 0.5$ | 74.13 |
| w/ full-capacity | $73.5 \pm 0.9$ | $\mathbf{84.9 \pm 1.3}$ | $\mathbf{79.4 \pm 0.9}$ | $\mathbf{78.8 \pm 0.5}$ | $\mathbf{85.2 \pm 0.4}$ | $\mathbf{61.2 \pm 1.0}$ | $\mathbf{76.9 \pm 0.1}$ | $\mathbf{64.9 \pm 0.2}$ | **75.60** |
| w/ 3-layer Student | $\mathbf{74.6 \pm 0.4}$ | $84.6 \pm 1.4$ | $76.4 \pm 1.0$ | $77.9 \pm 0.1$ | $82.7 \pm 1.1$ | $61.0 \pm 0.3$ | $75.0 \pm 0.1$ | $63.6 \pm 0.4$ | 74.48 |
| w/ 2-layer Student | $72.6 \pm 0.5$ | $81.4 \pm 0.4$ | $77.3 \pm 1.5$ | $77.6 \pm 0.2$ | $80.6 \pm 0.4$ | $60.8 \pm 0.4$ | $74.7 \pm 0.4$ | $63.0 \pm 0.1$ | 73.50 |

better than TGCL-DLSA (w/D-SLA) even though D-SLA outperformed GraphLog. Therefore, a better teacher does not necessarily produce better distillation for the student, as previously observed and analyzed in supervised learning (Menon et al., 2021; Kaplun et al., 2022; Zong et al., 2023).

We also observe that TGCL-NTXent produces poor performance compared to TGCL-DSLA and often even compared to its teacher models. This is because NT-Xent loss focuses only on contrasting the global representations, failing to capture the local graph characteristics. Therefore, even when a regularized notion of both local and global semantics is provided in terms of the distilled distance, NT-Xent-based TGCL models failed to utilize it efficiently for the link prediction tasks. We can also validate this hypothesis by comparing the performance of GraphCL, which uses NT-Xent loss, compared to the other methods (Table 2).

## 4.3 Experiments with Compressed Student

Table 3 demonstrates the performance of a student TGCL-DSLA model on the downstream molecular property prediction task. We can see that, with the same capacity (i.e., 5 layers of GNN) as the teacher module of D-SLA, our proposed student network consistently outperformed the teacher. As we decrease the capacity of our student network by reducing the number of layers, the overall performance reduces. However, we observe that even with 3 layers of GNN, our student module outperforms the teacher D-SLA model. Therefore, these results demonstrate that our proposed TGCL framework can compress the student representation network by enabling smoothened knowledge transfer from a pre-trained teacher to the student representation learning model.

## 5 Conclusion & Discussion

We utilize Knowledge distillation (KD) for graph representation learning, where a self-supervised pre-trained teacher model is used to guide the training of a student model to obtain more generalized representations. Extensive experimentation demonstrates the effectiveness of the proposed method in improving the performance of graph classification and link prediction tasks. However, there are still many open challenges in graph representation learning, such as the efficient handling of large-scale graphs, the ability to handle heterogeneity and multimodality, and the development of robust methods for noisy or incomplete data. Probing these challenges further and developing new graph representation learning techniques are in the scope of future research direction.

***Limitations.*** The performance of a student network heavily depends on the teacher's quality. While a more accurate 'teacher' does not necessarily lead to better distillation, a 'bad' teacher can reduce the student's performance. In particular, a better teacher yields a better approximation of the Bayes class probability distribution while leading to higher variance (i.e., unstable predictions) (Menon et al., 2021). Further, the teacher-student architecture is computationally expensive as we first need to train the teacher, followed by the student.

***Broader Impact.*** KD can significantly impact graph representations, with broader implications for various fields including bioinformatics, drug discovery, social network analysis, recommendation systems, etc. A few potential impacts of our work are as follows:

(a) Improves the efficiency and scalability of graph representation learning by enabling "soft" knowledge transfer from a pre-trained teacher model to a smaller, more efficient student network.

(b) Improves the generalization performance of graph representation learning models by leveraging the 'dark knowledge' encoded in a pre-trained teacher model's representations.

Overall, KD has the potential to significantly impact graph representations, therefore on various applications that rely on graphs and network analysis.

***Reproducibility Statement.*** To ensure that the proposed work is reproducible, we have included an Algorithm (Refer to Appendix Algorithm 1). We have clearly defined the loss functions in Section 3.3. The implementation details and hyperparameters are specified in Section D. The code of the proposed method is available at: `https://anonymous.4open.science/r/TGCL-400E/`. This is an anonymous link that doesn't reveal the author's identity. We have also included ReadMe files (inside each folder) to reproduce our results for convenience.

***Ethics statement.*** The ideas and techniques proposed in this paper can be useful in several real-world applications including the medical domain and e-commerce applications. We have touched on both the theoretical as well as experimental aspects of this problem in our work. We believe our results/findings should be available for all scientific communities for further research and development in this area, independent of their background (e.g., race, caste, creed, gender, nationality, etc.). The datasets used in our experiments are purely academic, and we do not think our work poses any specific ethical questions or creates potential biases against any particular groups.

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

# A    APPENDIX

# B    THEORETICAL DISCUSSION

**Proposition B.1.** *[Gutmann & Hyvärinen (2010); Section 2.1] Noise contrastive estimation of the density of a given random variable is equivalent to binary classification between the samples from that distribution and samples drawn from another arbitrary noisy distribution.*

**Proposition B.2.** *[Menon et al. (2021); Lemma 1] In supervised learning, the variance of Bayesian-distilled risk, obtained using the soft labels, weighted by the likelihood from the Bayes teacher in the form of knowledge distillation (KD), remains lower than the variance of empirical risk, obtained using the 'hard' class-labels, i.e.,*

$$\mathbb{V}_{\overline{\mathcal{G}} \sim \mathcal{G}}[\hat{R}_*(f, \overline{\mathcal{G}})] \leq \mathbb{V}_{\overline{\mathcal{G}} \sim \mathcal{G}}[\hat{R}(f, \overline{\mathcal{G}})] \tag{13}$$

*where, $f$ is a GNN model. $\mathbb{V}[\cdot]$ denotes the variance of a random variable. $\overline{\mathcal{G}}$ denotes the input graph dataset, sampled from the space of graphs, $\mathcal{G}$. $|\overline{\mathcal{G}}|$ denots the cardinality of $\overline{\mathcal{G}}$. $\boldsymbol{e}_{G_i}$ are the hard labels for input graph $G_i$ and $\boldsymbol{p}^*(G_i)$ is the Bayes class-probability distribution, predicted using the Bayes teacher. $\hat{R}_*(f, \overline{\mathcal{G}})$ and $\hat{R}(f, \overline{\mathcal{G}})$ denotes the Bayesian-distilled risk and the empirical risk respectively and defined as follows:*

$$\hat{R}_*(f, \overline{\mathcal{G}}) = \frac{1}{|\overline{\mathcal{G}}|} \sum_{G \in \overline{\mathcal{G}}} \boldsymbol{p}^*(G)^T \ell(f(G)) \tag{14}$$

$$\hat{R}(f, \overline{\mathcal{G}}) = \frac{1}{|\overline{\mathcal{G}}|} \sum_{G \in \overline{\mathcal{G}}}^{|\overline{\mathcal{G}}|} \boldsymbol{e}_G^T \ell(f(G)) \tag{15}$$

*$\ell$ is the empirical loss on $f$ (e.g., softmax cross-entropy). The equality holds iff $\forall G \in \mathcal{G}$, the loss values $\ell(f(G))$ are constant on the support of $\boldsymbol{p}^*(G)$.*

**Proposition B.3.** *[Maurer & Pontil (2009); Theorem 6 & Menon et al. (2021); Proposition 3] For any $\delta \in (0, 1)$ with probability at least $1 - \delta$ over the sample space, the reduction of variance for Bayesian-distilled risk, $\hat{R}_*(f, G)$ upper-bounds the population risk, $R(f)$, leading towards better model generalization, as follows:*

$$R(f) - \hat{R}_*(f, G) \leq \mathcal{O}\left(\sqrt{\mathbb{V}_{|\overline{\mathcal{G}}|}^*(f)/|\overline{\mathcal{G}}|} \cdot \sqrt{\log(\mathcal{M}_{|\overline{\mathcal{G}}|}^*} + \log(\mathcal{M}_{|\overline{\mathcal{G}}|}^*/\delta)/|\overline{\mathcal{G}}|)\right) \tag{16}$$

*where, $f$ is the GNN model from a fixed hypothesis class, with induced class $\mathcal{H}^* \subset [0, 1]^{\mathcal{G}}$ of functions $\langle(G) = \boldsymbol{p}^*(G)^T \ell(f(G))$ and $\mathcal{H}^*$ has uniform covering number $\mathcal{N}_\infty$. $\ell$ is a bounded loss (which can be enforced in practice using regularization). $\mathcal{M}_{|\overline{\mathcal{G}}|}^* = \mathcal{N}_\infty(\frac{1}{|\overline{\mathcal{G}}|}, \mathcal{H}^*, 2|\overline{\mathcal{G}}|)$ and $\mathbb{V}_{|\overline{\mathcal{G}}|}^*$ is the empirical variance of $\{\boldsymbol{p}^*(G_i)^T \ell(f(G_i))\}_{i \in \overline{\mathcal{G}}}$.*

# C    METHOD OVERVIEW & PSEUDO-CODE

In this section, we present the pseudo-code in Algorithm 1.

# D    EXPERIMENTAL SETUP

## D.1    MOLECULAR GRAPH CLASSIFICATION

This section presents the implementation details and dataset descriptions of our experiments on molecular graph classification and link prediction tasks. For all experiments, we use PyTorch (Paszke et al., 2019) and PyTorch Geometric libraries (Fey & Lenssen, 2019) with a single NVIDIA A30 Tensor Core GPU for all of our experiments.

---

**Algorithm 1:** PROPOSED TEACHER-GUIDED GRAPH LEARNING (TGCL-DSLA)

---

**Input:** $f_{teacher}$: Teacher model, $\mathcal{D}_{train}$: Training set of graphs (unlabelled)
**Output:** $f_s$: Student model

1 **for** *sampled mini-batch,* $\mathcal{G}_B := \{G_i\}_i \in \mathcal{D}_{train}$ **do**

    /* Create multiple perturbations for each (anchor) graph.      */

2    $\{G_{p_{ij}}\}_j \xleftarrow[variations]{perturbed} G_i$

    /* Compute *distilled perception distances* from $f_{teacher}$.      */

3    $D(G_i, G_{p_{ij}}) \;\forall i, j$ between anchor, $G_i$ and the perturbations, $G_{p_{ij}}$

4    $D(G_i, G_{n_{ij}}) \;\forall i, j$ where $G_{n_{ij}} \in \mathcal{G}_B - G_i$

    /* Obtain student's representations      */

5    Obtain representations for anchor graphs, $f_s(G_i)$ and the perturbations, $f_s(G_{p_{ij}})$

    /* Compute loss functions.      */

6    Compute $\sum_i \mathcal{L}_{T-soft}(G_i)$ among the anchor graphs, $G_i$ and its perturbations, $G_{p_{ij}}$

7    Compute $\sum_i \mathcal{L}_{T-percept}(G_i)$ among the anchor graphs, $G_i$ and its perturbations, $G_{p_{ij}}$

8    Compute $\sum_i \mathcal{L}_{T-Margin}(G_i)$ among the anchor graphs, $G_i$, its perturbations, $G_{p_{ij}}$ and negative instances, $G_{n_{ij}}$.

9    Minimize total mini-batch loss to update student $f_s$:
    $\sum_i \mathcal{L}_{T-soft}(G_i) + \lambda_1 \sum_i \mathcal{L}_{T-percept}(G_i) + \lambda_2 \sum_i \mathcal{L}_{T-Margin}(G_i)$

10 Return $f_s$

---

**Datasets.** For our first experiments on molecular graph classification, we use the ZINC dataset (Sterling & Irwin, 2015), a large corpus of 2 million unlabelled molecules for pretraining the teacher and student network. For the downstream tasks, we experimented with 8 labeled molecular datasets from MolecularNet (Wu et al., 2018). The molecule classes are determined using the biophysical and physiological properties. In Table 4, we present the details of these molecular datasets used for our experiments.

**Implementation details.** For our proposed framework, we use the same network architecture for both the teacher and the student model. In particular, we use Graph Isomorphism Networks (GINs) (Xu et al., 2019) as applied in the previous works Hu* et al. (2020); Xu et al. (2021); Kim et al. (2022). These networks consist of 5 layers with 300 dimensional embeddings for nodes and edges along with average pooling strategies for obtaining the graph representations. To obtain distilled perception distance from the teacher network, we use global average pooling to extract the fixed-length features from each layer.

We use the official D-SLA codes[1] provided by Kim et al. (2022) as the backbone for our experiments and apply the same perturbation strategies as used in (Kim et al., 2022). In particular, their perturbation strategy aims to minimize the risk of creating unreasonable cycles, reducing the chance of significant change in the chemical properties. For our experiments, we use three perturbations for each input sample.

We report results using two different teacher modules, trained using existing GraphLog (Xu et al., 2021) and D-SLA (Kim et al., 2022) while training the following student network using the loss functions as proposed in Section 3.3. We divide the perceptual distances by 4 and 1 as we use GraphLog (Xu et al., 2021) and D-SLA (Kim et al., 2022) as the teacher, respectively. For TGCL-

Table 4: Descriptions of Molecular datasets.

| Datasets | #Compounds | Tasks |
|---|---|---|
| ZINC15 | 2,000,000 | - |
| BBBP | 2,039 | 1 |
| ClinTox | 1,478 | 2 |
| MUV | 93,087 | 17 |
| HIV | 41,127 | 1 |
| BACE | 1,513 | 1 |
| SIDER | 1,427 | 27 |
| Tox21 | 7,831 | 12 |
| ToxCast | 8,575 | 617 |

NTXent, we use $\tau = 10$ in Equation 4. For TGCL-DSLA, we use $\lambda_1$ and $\lambda_2$ to 1.0 and 0.5 respectively for the student model. For $\mathcal{L}_{T-soft}$ loss, we set the temperature, $\tau = 10$ (Equation 6) and $\alpha = 0.95$ (Equation 8). For $\mathcal{L}_{T-margin}$, we set $\beta = 5$. Both teacher and student models are trained

---

[1]https://github.com/dongkikim95/d-sla

Table 5: Impact of individual loss components. (at $\alpha = 0.95$, $\tau = 10$)

| $\mathcal{L}_{T-soft}$ | $\mathcal{L}_{T-percept}$ | $\mathcal{L}_{T-margin}$ | BBBP | ClinTox | MUV | HIV | BACE | SIDER | Tox21 | ToxCast | Avg |
|---|---|---|---|---|---|---|---|---|---|---|---|
| ✓ | ✗ | ✗ | $72.9 \pm 1.4$ | $79.8 \pm 1.2$ | $79.1 \pm 0.7$ | $77.7 \pm 0.6$ | $81.9 \pm 0.3$ | $62.1 \pm 0.7$ | $\mathbf{76.9 \pm 0.2}$ | $64.1 \pm 0.3$ | 74.31 |
| ✗ | ✓ | ✗ | $71.6 \pm 0.8$ | $74.5 \pm 0.7$ | $76.6 \pm 1.3$ | $78.5 \pm 1.1$ | $81.7 \pm 0.9$ | $61.7 \pm 0.6$ | $75.7 \pm 0.6$ | $62.9 \pm 0.3$ | 72.90 |
| ✗ | ✗ | ✓ | $72.7 \pm 0.6$ | $77.9 \pm 2.0$ | $74.1 \pm 0.9$ | $76.6 \pm 0.4$ | $82.9 \pm 0.6$ | $\mathbf{62.8 \pm 0.5}$ | $74.2 \pm 0.1$ | $61.9 \pm 0.8$ | 72.89 |
| ✗ | ✓ | ✓ | $\mathbf{73.6 \pm 0.5}$ | $81.2 \pm 1.1$ | $75.7 \pm 0.4$ | $77.3 \pm 1.4$ | $83.2 \pm 0.3$ | $\mathbf{62.8 \pm 0.6}$ | $75.2 \pm 0.2$ | $63.3 \pm 0.5$ | 74.04 |
| ✓ | ✓ | ✗ | $72.8 \pm 0.1$ | $81.6 \pm 0.5$ | $79.2 \pm 0.5$ | $\mathbf{78.8 \pm 0.9}$ | $81.4 \pm 1.2$ | $59.7 \pm 0.5$ | $76.3 \pm 0.2$ | $63.8 \pm 0.1$ | 74.20 |
| ✓ | ✗ | ✓ | $72.1 \pm 0.5$ | $84.0 \pm 2.3$ | $76.7 \pm 1.3$ | $77.9 \pm 0.7$ | $82.5 \pm 0.5$ | $61.4 \pm 0.3$ | $76.3 \pm 0.2$ | $64.3 \pm 0.7$ | 74.40 |
| ✓ | ✓ | ✓ | $73.5 \pm 0.9$ | $\mathbf{84.9 \pm 1.3}$ | $79.4 \pm 0.9$ | $78.8 \pm 0.5$ | $\mathbf{85.2 \pm 0.4}$ | $61.2 \pm 1.0$ | $\mathbf{76.9 \pm 0.1}$ | $\mathbf{64.9 \pm 0.2}$ | **75.60** |

using batch-size of 256 and for 25 epochs with learning rate $1e - 3$ and Adam optimizer (Kingma & Ba, 2014).

## D.2 LINK PREDICTION

**Datasets.** For this task, we select three datasets i.e., COLLAB, IMDB-Binary, and IMDB-Multi from the TU dataset benchmark Morris et al. (2020).

COLLAB is a dataset of scientific collaboration networks. It contains $4,320$ graphs where the researcher and their collaborators are nodes and an edge indicates collaboration between two researchers. A researcher's ego network has three possible labels: *High Energy Physics*, *Condensed Matter Physics*, and *Astro Physics*, representing the field of the researchers.

IMDB-Binary is a movie collaboration dataset. It consists of the ego networks of actors/actresses from the movies in IMDB. It consists of $2,039$ graphs. For each graph, the nodes are actors/actresses, with an edge between them if they appear in the same movie. These graphs are derived from the Action and Romance genres.

IMDB-Multi is a relational dataset that consists of a network of actors or actresses, played in movies in IMDB. It contains 1,478 graphs. As before, a node represents an actor or actress, and an edge connects two nodes when they appear in the same movie. The edges are collected from three different genres: Comedy, Romance, and Sci-Fi.

**Implementation details.** For our experiments, we use Graph Convolutional Network (GCN) Kipf & Welling (2017) for both teacher and student models. These networks consist of three layers with 300 dimensions for embeddings. As before, we use the same perturbation strategy as applied in Kim et al. (2022). We have also experimented with two different teachers, *i.e.*, D-SLA Kim et al. (2022) and GraphLog Xu et al. (2021). We use a batch size of 32 and a learning rate of 0.001 for training the student representation learning models. For TGCL-NTXent, we set $\tau = 10$ (Eq. 4). For TGCL-DSLA, we use $\lambda_1$ and $\lambda_2$ to 0.7 and 0.0, respectively. For $\mathcal{L}_{T-soft}$ loss, we select the temperature, $\tau$ from three different values i.e., $\{5, 10, 20\}$ (Equation 6) and set $\alpha = 0.95$ (Eq. 8).

## E ADDITIONAL EXPERIMENTS

In this section, we present additional experiments to study the effects of different hyper-parameters and loss components for our TGCL-DSLA framework.

**Impact of different loss components.** In Table 5, we first demonstrate the impact of different loss components. The first three rows demonstrate the performance of individual loss components. We observe that $\mathcal{L}_{soft}$ is the most important component, providing the maximum performance boost for the downstream molecular prediction tasks. The other two loss components, i.e. $\mathcal{L}_{T-percept}$ and $\mathcal{L}_{T-margin}$ act as regularizer. While, individually they do not perform well, incorporating them with $\mathcal{L}_{soft}$ in $\mathcal{L}_{overall}$, we observe a significant boost in the overall performance.

**Impact of hyper-parameters of $\mathcal{L}_{T-soft}$.** Since $\mathcal{L}_{T-soft}$ is the most important loss component, we further analyze hyper-parameters associated with it. We can see in Eq. 8, $\mathcal{L}_{soft}$ is similar to the distillation loss for classification tasks, consisting of two loss components i.e. $\mathcal{L}_{KD}$ (Eq. 6) and $\mathcal{L}_{wGD}$ (Eq. 7). Here, we analyze the temperature term, $\tau$ for $\mathcal{L}_{KD}$ followed by the weights of these components, $\alpha$.

Table 6: Impact of the temperature, $\tau$. (at $\alpha = 0.95$)

| $\tau$ | BBBP | ClinTox | MUV | HIV | BACE | SIDER | Tox21 | ToxCast | Avg |
|---|---|---|---|---|---|---|---|---|---|
| 1 | **75.1 ± 0.4** | **86.7 ± 1.6** | 74.4 ± 0.1 | 77.5 ± 0.6 | 83.3 ± 1.0 | 61.2 ± 0.4 | 75.6 ± 0.1 | 63.4 ± 0.4 | 74.65 |
| 5 | 73.4 ± 0.2 | 81.8 ± 1.4 | 77.3 ± 2.3 | 78.6 ± 0.6 | 83.8 ± 0.8 | 61.7 ± 0.8 | 76.5 ± 0.3 | 63.9 ± 0.4 | 74.63 |
| 10 | 73.5 ± 0.9 | 84.9 ± 1.3 | **79.4 ± 0.9** | **78.8 ± 0.5** | **85.2 ± 0.4** | 61.2 ± 1.0 | **76.9 ± 0.1** | **64.9 ± 0.2** | **75.60** |
| 20 | 72.9 ± 0.6 | 83.9 ± 2.6 | 77.1 ± 0.5 | 78.3 ± 0.7 | 84.0 ± 0.8 | 61.8 ± 0.4 | 76.2 ± 0.4 | 64.6 ± 0.5 | 74.85 |
| 100 | 73.6 ± 0.1 | 80.6 ± 0.2 | 76.8 ± 2.8 | 78.7 ± 1.1 | 84.0 ± 0.5 | **62.3 ± 0.5** | 76.1 ± 0.3 | 64.5 ± 0.2 | 74.58 |

Table 7: Impact of $\alpha$. (at $\tau = 10$)

| $\alpha$ | BBBP | ClinTox | MUV | HIV | BACE | SIDER | Tox21 | ToxCast | Avg |
|---|---|---|---|---|---|---|---|---|---|
| 0 | 73.2 ± 0.7 | 80.2 ± 1.8 | 76.4 ± 0.7 | 78.1 ± 0.6 | 84.1 ± 0.9 | 62.3 ± 0.5 | 75.5 ± 0.3 | 63.8 ± 0.3 | 74.20 |
| 0.5 | **73.6 ± 0.7** | 82.5 ± 1.2 | 75.0 ± 1.5 | 78.4 ± 0.6 | **85.6 ± 0.5** | **62.4 ± 0.1** | 75.8 ± 0.1 | 64.5 ± 0.3 | 74.73 |
| 0.95 | 73.5 ± 0.9 | **84.9 ± 1.3** | **79.4 ± 0.9** | **78.8 ± 0.5** | 85.2 ± 0.4 | 61.2 ± 1.0 | **76.9 ± 0.1** | **64.9 ± 0.2** | **75.60** |
| 1.0 | 72.8 ± 0.6 | 83.6 ± 1.2 | 77.6 ± 1.6 | **78.8 ± 0.4** | 83.4 ± 1.3 | 61.3 ± 0.6 | 76.6 ± 0.3 | 64.1 ± 0.2 | 74.78 |

Table 8: Sensitivity analysis of $\lambda_1$ and $\lambda_2$ for $TGCL - DSLA(w/DSLA)$ model.

| | $\lambda_1 = 0.3$ | $\lambda_1 = 0.5$ | $\lambda_1 = 0.7$ | $\lambda_1 = 1.0$ |
|---|---|---|---|---|
| BBBP | 72.55 ± 0.32 | 72.03 ± 0.59 | **74.46 ± 1.54** | 73.5 ± 0.9 |
| ClinTox | 81.60 ± 0.21 | 80.33 ± 2.39 | 82.95 ± 0.75 | **84.9 ± 1.3** |
| BACE | 83.64 ± 0.71 | 83.18 ± 1.02 | 83.38 ± 0.33 | **85.2 ± 0.4** |
| MUV | 76.28 ± 0.91 | 76.00 ± 0.37 | 77.08 ± 1.45 | **79.4 ± 0.9** |
| HIV | 78.61 ± 0.62 | **78.93 ± 0.58** | 78.64 ± 0.46 | 78.8 ± 0.5 |

| | $\lambda_2 = 0.3$ | $\lambda_2 = 0.5$ | $\lambda_2 = 0.7$ | $\lambda_2 = 1.0$ |
|---|---|---|---|---|
| BBBP | 72.96 ± 0.27 | 73.5 ± 0.9 | 72.85 ± 0.24 | **74.12 ± 0.41** |
| ClinTox | 83.53 ± 1.51 | **84.9 ± 1.3** | 81.95 ± 0.93 | 80.83 ± 1.68 |
| BACE | 83.33 ± 0.31 | **85.2 ± 0.4** | 83.60 ± 0.90 | 83.53 ± 0.72 |
| MUV | 75.79 ± 0.78 | **79.4 ± 0.9** | 76.75 ± 1.54 | 76.02 ± 0.11 |
| HIV | 78.66 ± 0.81 | 78.8 ± 0.5 | 78.17 ± 0.62 | **79.49 ± 0.84** |

In Table 6, we demonstrate the results by varying the temperature, $\tau$ to $\{1, 5, 10, 20, 100\}$. We observe that at a lower temperature of $\tau = 1$, we achieve the best performance for BBBP and Clintox dataset, while the performance remains lower for the other datasets. On the other hand, at $\tau = 100$, we achieve the best performance for SIDER. Finally, we obtain the most consistent result as we select $\tau = 10$ and set it to report our results.

Next, in Table 7, we analyze the impact of $\alpha$ with fixed $\tau = 10$. A larger value of $\tau$ provides more weight to $\mathcal{L}_{KD}$. We can see that as we increase $\alpha$ to a non-zero value, it improves the overall performance of the model. However, as we choose $\alpha = 1$ to entirely remove $\mathcal{L}_{wGD}$, the performance tends to reduce. We achieve the best performance at $\alpha = 0.95$.

**Sensitivity analysis of $\lambda_1$ and $\lambda_2$ for $TGCL - DSLA(w/DSLA)$ model.** In Table 8, we present the performance of $TGCL - DSLA(w/DSLA)$ model as we vary $\lambda_1$ and $\lambda_2$ in Eq. 12. We first vary $\lambda_1$ to $\{0.3, 0.5, 0.7, 1.0\}$ as we fix $\lambda_2 = 0.5$. We observe that the performance improves as we choose larger values, *i.e.*, when we set $\lambda_1$ to 0.7 or 1.0. Next, we vary $\lambda_2$ to $\{0.3, 0.5, 0.7, 1.0\}$ as we fix $\lambda_1 = 1.0$. Here, we observe that we achieve the average performance as we set $\lambda_2 = 0.5$

