# OpenReview forum: "A Teacher-Guided Framework for Graph Representation Learning"
_ICLR.cc/2024/Conference — ICLR 2024 Conference Withdrawn Submission_

### Official Review · Reviewer_RhAx · 2023-10-31

**Soundness:** 2 fair
**Presentation:** 3 good
**Contribution:** 2 fair
**Rating:** 3
**Confidence:** 4

**Summary:**

This paper proposes a contrastive learning method for graph representation learning, where a self-supervised pre-trained teacher model is used to guide the training of a student model to obtain more generalized representations. Extensive experimentation demonstrates the effectiveness of the proposed method in improving the performance of graph classification and link prediction tasks.

**Strengths:**

1. The idea of using teacher-student architecture in graph representation learning is interesting;

2. The authors designed several loss functions to make use of the soft labels learned from the teacher model;

3. The authors did comprehensive experiments on link prediction and classification tasks;

4. This paper is well written and easy to follow.

**Weaknesses:**

1. The major concern is the technical soundness of the teacher-student architecture. Intuitively, if you have a perfect teacher model, you can use it directly to calculate graph embeddings. Is it necessary to design such complicated contrastive learning losses to distill from the teacher model? The teacher-student model is originally designed to distill knowledge from large models and inject into small models, however, in this paper, the purpose of teacher-student is not to reduce the model size.

2. The performance of the student model is theoretically bounded by the teacher model. I'm curious why the propose model TGCL-DSLA can perform better than the teacher DSLA and GraphLoG?

3. For graph classification task, the datasets are all on molecule properties, which is quite limited. It is better to add more types of graph classification tasks to see how the proposed model performs on various type of graphs.

4. From Table 1, most experimental results do not show significant improvement over baselines.

4. According to Figure 4, why the proposed TGCL-DSLA learns the best graph embedding? It is hard to see the difference between (a) and (c).

**Questions:**

See weaknesses above.

---

> ### Author Response · Authors · 2023-11-13
> **Rebuttal by Authors**
>
> We thank the reviewer for providing their valuable feedback for our paper.
>
> **Weakness 1: The major concern is the technical soundness of the teacher-student architecture. Intuitively, if you have a perfect teacher model, you can use it directly to calculate graph embeddings**
>
> Yes. A perfect teacher model can produce perfect embeddings. However, as mentioned in our introduction (and in Figure 1), the existing techniques do not produce such a perfect model.
>
> While the teacher-student model is originally designed to distill knowledge from large models and inject into small models, several recent advancements are proposed and shown that the applications of KD technique is not only limited to reducing the size of the models
>
> **Weakness 2: The performance of the student model is theoretically bounded by the teacher model. I'm curious why the proposed model TGCL-DSLA can perform better than the teacher DSLA and GraphLoG?**
>
> Please note that *“performance of the student model is theoretically bounded by the teacher model”* is **incorrect**. We would like to request the reviewer to provide a reference for this claim so that we can make further comments.
>
> Also, we would like to point out that Menon et al. [1] theoretically proved that – *“Bayes teacher” providing the true class probabilities can lower the variance of the student objective, and thus improve performance.”* [1]. (also refer to our Appendix B for detailed discussions on this.)
>
> Several studies also empirically verified that. For example, quoting Furlanello et al. [2] – *“We study KD from a new perspective: rather than compressing models, we train students parameterized identically to their teachers. Surprisingly, these Born-Again Networks (BANs), outperform their teachers significantly, both on computer vision and language modelling tasks.”*
>
> The argument “performance (or generalization capacity) of the student model is usually upper bounded by the teacher model” is only empirically valid when the capacity of the student model is lower than the teacher (see our experiments in Section 4.3).
> Key differences: We highlighted the key differences between our paper and the existing works in Section 2.2
>
> [1] A Statistical Perspective on Distillation (ICML 2021)
>
> [2] Born-Again Neural Networks (ICML 2018)
>
> **Weakness 3: For graph classification task, the datasets are all on molecule properties, which is quite limited. It is better to add more types of graph classification tasks to see how the proposed model performs on various type of graphs.**
>
> We followed the previous existing papers to set up our experiments [1,2,3].
> We have shown results on general/social network graphs such as COLLAB, IMDB-Binary, and IMDB-Multi (see Table 2). We have achieved up to $\approx$ 5% improvements using our proposed method.
>
> [1] Graph contrastive learning automated. (ICML 2021)
>
> [2] Self-supervised graph-level representation learning with local and global structure (ICML 2021)
>
> [3]  Graph self-supervised learning with accurate discrepancy learning. (NeurIPS, 2022)
>
> **Weakness 4: From Table 1, most experimental results do not show significant improvement over baselines.**
>
> Please note that even a 1% improvement on molecular property prediction datasets is significant. For example, DSLA (accepted at NeurIPS’22) outperformed GraphLoG (ICML’21) by <1%. In comparison, we improved the performance by $\approx$ 1.6%.
> Further, in Table 2, we outperformed the existing models by up to $\approx$ 5% on general, social network graphs.
>
> **Weakness 5: According to Figure 4, why the proposed TGCL-DSLA learns the best graph embedding? It is hard to see the difference between (a) and (c).**
>
> Yes. Visually Figure (a) and (c) produces similar t-SNE plots and it is not easy to determine the best graph embeddings. We have updated our paper accordingly.

---

### Official Review · Reviewer_xgHF · 2023-10-31

**Soundness:** 3 good
**Presentation:** 3 good
**Contribution:** 2 fair
**Rating:** 3
**Confidence:** 4

**Summary:**

This paper proposed a contrastive learning (CL) based knowledge distillation (KD) framework for Graph Neural Networks (GNNs).  Particularly, the authors incorporated `soft labels` to facilitate a more regularized discrimination. In the teacher-student KD framework, the student network learns the representation by distilling the representations produced by the teacher network trained using unlabelled graphs.

**Strengths:**

1. The overall written is clear and easy to follow.
2. The experiments show that the distilled results perform better than the original teachers.

**Weaknesses:**

I'm not the expert in contrastive learning. But as far as I know, the soft labeled CL is not novel. For example, "Soft-Labeled Contrastive Pre-training for Function-level Code Representation" in EMNLP 2022 has already proposed to leverage soft labels to mitigate similar issues of hard labels, .e.g, semantic problem, false negative case. I'm not sure whether other related works also applied the similar soft labeled CL framework.

**Questions:**

In the experiments, the student model can consistently outperform the teacher model via the distillation. What if you use the current student model as the next round teacher model, can your performance continue to increase? or how many rounds KD will make the improvement negligible?

---

> ### Author Response · Authors · 2023-11-13
> **Rebuttal by Authors**
>
> We thank the reviewer for providing their valuable feedback for our paper.
>
> **Weaknesses:**
> *I'm not the expert in contrastive learning. But as far as I know, the soft labeled CL is not novel. For example, "Soft-Labeled Contrastive Pre-training for Function-level Code Representation" in EMNLP 2022 has already proposed to leverage soft labels to mitigate similar issues of hard labels, .e.g, semantic problem, false negative case. I'm not sure whether other related works also applied the similar soft labeled CL framework.*
>
>
> **Ans.**
> Please note that soft-labelled CL (or the idea of incorporating KD with CL) is not new, and several papers are proposed and several advancements are made in the subsequent works. For example, **SimCLR-V2** [1] was among the first papers to incorporate KD with CL. However, several follow-up works were proposed and advanced the literature, and well accepted in different Tier-1 conferences.
>
> In Section 2.2, we have systematically reviewed the existing works and provided a detailed discussion identifying the key differences between our proposed method compared to the existing works.
>
> [1] Big self-supervised models are strong semi-supervised learners (NeurIPS 2020)
>
> **Questions:**
> *In the experiments, the student model can consistently outperform the teacher model via the distillation. What if you use the current student model as the next round teacher model, can your performance continue to increase? or how many rounds KD will make the improvement negligible?*
>
>
> An iterative evolutionary paradigm does not help the model performance. The reasons are as follows:
>
> - A better teacher model does not necessarily produce a better student model. [1,2,3]
> - A teacher model already provides an estimation of the Bayes class probability distribution over the labels.
>
> Section 4.1 of [1] theoretically explained the Bias-Variance Bound trade-off for Distillation from an imperfect teacher. The Bias-Variance Bound trade-off explains why our KD-based TGCL framework does not need a perfect model. Therefore, a student model typically achieves the best performance after training using the first-level teacher model. The performance saturates in their subsequent iterations.
>
> [1] A Statistical Perspective on Distillation (ICML 2021)
>
> [2] Knowledge distillation: Bad models can be good role models. (NeurIPS 2022)
>
> [3]  Better teacher better student: Dynamic prior knowledge for knowledge distillation (ICLR 2023)

---

### Official Review · Reviewer_2WDw · 2023-11-05

**Soundness:** 3 good
**Presentation:** 3 good
**Contribution:** 3 good
**Rating:** 5
**Confidence:** 3

**Summary:**

**Summary**

The paper proposed an distillation-based unsupervised representation learning method. Given a teacher model pretrained on an unlabel dataset, a student model is pretrained on the same pretrained dataset with the TGCL objective, and then further finetuned on the downstream datasets.

**Contributions**
 1. The authors modified existing contrastive learning objectives (e.g. NTXent, D-SLA) using the distilled perceptual distance.
 - In NTXent, they reweight the cosine similarity with the proposed distilled perceptual distance.
 - In D-SLA, they replace the graph edit distance with the proposed distilled perceptual distance

 2. This approach is implemented and evaluated on various node and graph classification datasets.

**Strengths:**

The paper is well-organized and well-written. The motivation and methodology is clearly stated.
Addressing my concerns would be greatly appreciated, and it could lead to an increase in my rating.

**Weaknesses:**

**Concern 1. Why can the student model trained by TGCL outperform the teacher model on the downstream tasks?**

TGCL primarily relies on the semantic similarity (soft label) provided by the teacher model in a knowledge distillation manner. In general distillation settings (e.g. model compression via distillation), the performance (or generalization capacity) of the student model is usually upper bounded by the teacher model.

Thus, I wonder why TGCL-student outperforms the teacher model? Which part of the algorithm contributes to the performance / information gain over the teacher model?



**Concern 2. Is TGCL sensitive to the pretraining method of the teacher model?**

As shown in Table 1, TGCL-analogues do not consistently reach the state-of-the-art (SOTA) performance. Does TGCL exclusively apply to teacher models trained using specific pretraining methods, such as GraphLog and D-SLA?

If TGCL can enhance teacher models trained by any algorithm, could it serve as an iterative evolutionary paradigm? For instance, a teacher model undergoes k epochs of pretraining, followed by k epochs of TGCL updates, and this process repeats until convergence.



**Concern 3: Does TGCL's performance depend on the quality of the teacher model?**

As TGCL heavily relies on the semantic similarity of teacher embeddings, I wonder to what extent the performance of TGCL relies on the representation quality of the teacher model

Intuitively, obtaining a proficient teacher model is as challenging as achieving high-quality unsupervised representations. If TGCL only works with a well-trained teacher models, it may not address unsupervised representation learning directly. Instead,  it serves as an incremental improvement to existing effective unsupervised pretraining methods.



**Concern 4: Does TGCL work when the teacher model overfits to the pretraining domain?**

If the teacher model fits the pretraining dataset well (or, ‘overfits the pretraining distribution’), it may encode domain-specific biases that hinder out-of-domain generalization. In this scenario, does it harm the downstream performance of the student model? What aspect of TGCL can help mitigate pretraining bias and enhance downstream generalization?

**Questions:**

My questions are stated in the 'weakness' section.

---

> ### Author Response · Authors · 2023-11-13
> **Rebuttal by Authors**
>
> We thank the reviewer for providing their valuable feedback for our paper.
>
> **Concern 1. Why can the student model trained by TGCL outperform the teacher model on the downstream tasks?**
>
> **Ans.**
> Please note that the “performance (or generalization capacity) of the student model is usually upper bounded by the teacher model” is *incorrect*. In fact, it has been theoretically proven that – *“Bayes teacher” providing the true class probabilities can lower the variance of the student objective, and thus improve performance.”* [1]. (also refer to our Appendix B for detailed discussions on this.)
>
> Several studies also empirically verified that. For example, quoting Furlanello et al. [2] – *“We study KD from a new perspective: rather than compressing models, we train students parameterized identically to their teachers. Surprisingly, these Born-Again Networks (BANs), outperform their teachers significantly, both on computer vision and language modeling tasks.”*
> The argument “performance (or generalization capacity) of the student model is usually upper bound by the teacher model” is only empirically valid when the capacity of the student model is lower than the teacher (see our experiments in Section 4.3).
>
>
> **Key differences:** We highlighted the key differences between our paper and the existing works in Section 2.2
>
> [1] A Statistical Perspective on Distillation (ICML 2021)
>
> [2] Born-Again Neural Networks (ICML 2018)
>
> **Concern 2. Is TGCL sensitive to the pretraining method of the teacher model?**
>
> **Concern 3: Does TGCL's performance depend on the quality of the teacher model?**
>
> **Ans.**
> Yes. Similar to the existing KD methods, TGCL is also sensitive to the pretraining method of the teacher model. However, it is not exclusively applied to teacher models trained using specific pretraining methods. Also, an iterative evolutionary paradigm does not help the model performance. The reasons are as follows:
>
> - A better teacher model does not necessarily produce a better student model.
> - A teacher model already provides an estimation of the Bayes class probability distribution over the labels.
>
> Section 4.1 of [1] theoretically explained the Bias-Variance Bound trade-off for Distillation from an imperfect teacher.
>
> **Answer to Concern 2:** In Table 1, our TGCL-DSLA consistently outperforms vanilla DSLA (teacher). Similarly, TGCL-GraphLoG consistently outperforms vanilla Graph-LoG (teacher).
>
> **Answer to Concern 2 and 3:** The Bias-Variance Bound trade-off explains why our KD-based TGCL framework does not need a perfect model. Therefore, a student model typically achieves the best performance after training using the first-level teacher model. Their performance saturates in their subsequent iterations.
>
> [1] A Statistical Perspective on Distillation (ICML 2021)
>
> **Concern 4: Does TGCL work when the teacher model overfits to the pretraining domain?**
>
> - Since we are dealing with unsupervised training for the teacher, in general, it should not overfit to a specific domain.
> - Even if the model gets overfitted to a specific domain, our TGCL framework still performs well as a KD-based student model does not require a perfect teacher.

---

> > ### Comment · Reviewer_2WDw · 2023-11-18
> > **Response to Authors' Answer to Concerns 1, 2 and 3**
> >
> > **Ans 1 to Concern 1.**
> >
> > Please note that the “performance (or generalization capacity) of the student model is usually upper bounded by the teacher model” is *incorrect*. ......
> >
> > **Response 1-1. Response to Ans 1 to Concern 1.**
> >
> > **Theoretical analysis would be helpful.**
> >
> > Thanks for the clarification. To my understanding, using the distilled perceptual distance can reduce the Bayes-distilled risk, which in this paper corresponds to the data-noise binary classification error in contrastive learning. However, there is still a gap between achieving an improved contrastive loss and ahieving an improved classification loss in the downstream task, i.e. the final graph / node classification tasks.
> >
> > It would be helpful if the authors can add some theoretical analysis that explains how to utilize the propositions in Appendix B to establish a generalization bound control for the final classification task using the TCGL method. This approach could be more illustrative than merely listing previous theoretical results without mathematical derivation, as it would clarify the necessary conditions for achieving improved generalization bound with the KD-based contrastive learning method.
> >
> > **Novelty of TCGL method on graph.**
> >
> > I appreciate that the authors have highlighted the key differences between this paper and previous works in Section 2.2. To my understanding, since the KD + CL paradigm is not new, the main contribution of this paper should lies on how is the proposed method exclusively tailored to graph data learning. While in the Limitations & Challenges section 2.2, the authors claim that TGCL can detect semantic changes between graphs with minor perturbations when compared to previous KD + CL methods, I believe this capability is primarily achieved through the use of the D-SLA framework (Section 3.3.2) and the D-SLA teacher, in which the graph edit distance plays an essential role.
> >
> > Although TGCL-DSLA and TGCL-GraphLoG is better than the vanilla teacher DSLA, and GraphLoG, the authors also show that TGCL-NTXent with DSLA teacher (e.g. the plain distilled perceptual loss) fails to outperform DSLA on two datasets. Does this means the DSLA framework plays a more decisive role in capturing graph features, than the proposed distilled perceptual loss?
> >
> > It would be helpful if the authors could provide more in-depth discussions on why the proposed distilled perceptual loss is specifically tailored to graph data learning. This additional explanation would help ensure the novelty of TCGL are not underestimated.
> >
> >
> >
> > **Ans to Concern 2 and Concern 3**
> >
> > Yes. Similar to the existing KD method ...
> >
> > **Response 2-2,3. Response to Ans to Concern 2 and Concern 3.**
> >
> > According to [1], a closer alignment of the imperfect teacher with the Bayes probability results in reduced variance, including the generalization bound, for the distilled student. Since the Bayes dsitill risk in this paper is the contrastive loss, rather than the downstream classification loss, the teacher model (e.g. DSLA) only provides an estimation of the Bayes probability over the data-loss binary classification probability, rather than the Bayes probability over the labels.
> >
> > As I have mentioned in the Response 1-1 above, a concrete theoretical analysis on how TCGL ahieves improved downstream performance would be helpful.
> >
> > As shown in the Table 2 of [2], a sequentially distilled model is able to outperform a one-shot distilled student. According to [1], this can be achieved when each time the distilled student (which is also the next teacher) is trained to be closed to the Bayes teacher. Given the results in [1] and [2], I can hardly draw the conclusion ‘iterative evolutionary paradigm does not help the model performance’ without empirical experiments. Instead, I am curious on
> >
> > 1. Why the current self-supervised teacher is able to provide Bayes probability over labels? There are no likelihood calibration on the training of self-supervised teacher.
> > 2. At each iteration, can we penalize the student and the teacher towards the Bayes teacher, to pursue a potential improvement in the successive distillation processes.
> >
> >
> >
> > [1] *Aditya K Menon, Ankit Singh Rawat, Sashank Reddi, Seungyeon Kim, and Sanjiv Kumar. A statistical perspective on distillation. In International Conference on Machine Learning, pp. 7632–7642. PMLR, 2021.*
> >
> > [2] *Tommaso Furlanello, Zachary Lipton, Michael Tschannen, Laurent Itti, and Anima Anandkumar. Born again neural networks. In International Conference on Machine Learning, pp. 1607–1616. PMLR, 2018.*

---

> > ### Comment · Reviewer_2WDw · 2023-11-18
> > **Response to Authors' Answer to Concern 4**
> >
> > **Ans to Concern 4**
> >
> > Since we are dealing with unsupervised training for the teacher ...
> >
> > **Response 3-4. Response to Ans to Concern 4.**
> >
> > Thanks for the clarification. I am convinced that the unsupervised teacher will not overfit to the specific domain. According to [1], the generalization bound improvement requires the teacher to be closed to a Bayes probability estimator. But I doubt on whether this requirement can be achieved by a unsupervised teacher or overfitted teacher.
> >
> > Without adding explicit probability calibration, how can the DSLA teacher provides a Bayes probability estimation on labels? According to Section 4.2, ‘Why can more accurate teachers distill worse’ in [1],  overfitted teacher is far worse than imperfect, as it provides accurate yet over-confident and poorly calibrated predictions. It would be highly appreciated if the authors can provide some theoretical or empirical analysis on this seeming contradiction.

---

> > > ### Author Response · Authors · 2023-11-20
> > > **Authors' response**
> > >
> > > We thank the reviewer for engaging in the discussion.
> > >
> > >
> > > **Response 1-1: “It would be helpful if the authors can add some theoretical analysis that explains how to utilize the propositions in Appendix B to establish a generalization bound control for the final classification task using the TCGL method.”**
> > >
> > >
> > > **Ans:-** Previous studies have already demonstrated that better representation learning using contrastive learning-based self-supervised techniques leads to better classification performance in the downstream tasks using theoretical analysis [1] as well as empirical results [2,3]. Therefore, for our work, it is sufficient to show that our proposed TGCL framework leads to better representations for the graphs. In other words, by efficiently optimizing the contrastive loss functions to obtain better-generalized representations. The theoretical results provided in Appendix B demonstrate it as follows:
> > >
> > >
> > > 1. While CL is an unsupervised learning task, its objective can be posed as a supervised learning task with artificially created pseudo-class labels (See Section 3.2: “The first result indicates…”). Therefore, it can provide an explicit probability calibration corresponding to these pseudo-class labels even though we are training using a self-supervised teacher.
> > > 2. Now, by using ‘soft labels’ in the form of Knowledge Distillation (KD), our proposed TGCL framework leads to better generalization by reducing the variance of Bayes-distilled risk corresponding to contrastive loss functions. Hence, we can achieve better-generalized representations that lead to improving the downstream classification tasks.
> > >
> > > Therefore, while our paper empirically demonstrates the efficiency of our proposed TGCL framework, we believe that the existing theoretical results are sufficient to show the better-generalized representations and hence, improved classification performance in the downstream tasks.
> > >
> > >
> > > [1] To Compress or Not to Compress - Self-Supervised Learning and Information Theory: A Review (Arxiv, 2023)
> > > [2] A Simple Framework for Contrastive Learning of Visual Representations (ICML, 2020)
> > > [3] Bootstrap your own latent: A new approach to self-supervised Learning (Arxiv 2020)
> > >
> > > **Response 2-2,3: Why the current self-supervised teacher is able to provide Bayes probability over labels? There are no likelihood calibration on the training of self-supervised teacher.**
> > >
> > >
> > > **Response 3-4: But I doubt on whether this requirement can be achieved by an unsupervised teacher or an overfitted teacher …Without adding explicit probability calibration, how can the DSLA teacher provides a Bayes probability estimation on labels?**
> > >
> > >
> > > **Ans:-** As explained earlier, while CL is an unsupervised learning task, its objective can be posed as a supervised learning task with artificially created pseudo-class labels (See Section 3.2: “The first result indicates…”). Therefore, it can provide an explicit probability calibration corresponding to these pseudo-class labels even though we are training using a self-supervised teacher.
> > >
> > >
> > > **Response 2-2,3: ... TGCL-NTXent with DSLA teacher (e.g. the plain distilled perceptual loss) fails to outperform DSLA on two datasets. Does this means the DSLA framework plays a more decisive role in capturing graph features, than the proposed distilled perceptual loss?**
> > >
> > >
> > > **Ans:-**
> > >  - Please note that the performance of the TGCL-NTXent with DSLA teacher remains within the error margin of the DSLA framework on the above-mentioned datasets (i.e., HIV and Tox21).
> > >  - Also, DSLA framework doesn't play a more decisive role as we achieve the best average performance using TGCL-NTXent with GraphLoG as the teacher.
> > >  - Finally, while we provide a generalized framework to accommodate knowledge distillation with the contrastive loss, it is not realistic to expect to achieve the best results with all possible combinations (no free lunch theorem).

---

> > > > ### Author Response · Authors · 2023-11-20
> > > > **Authors' response (contd.)**
> > > >
> > > > **Response 2-2,3: At each iteration, can we penalize the student and the teacher towards the Bayes teacher, to pursue a potential improvement in the successive distillation processes.**
> > > >
> > > >  **Response 2-2,3: I can hardly draw the conclusion ‘iterative evolutionary paradigm does not help the model performance’ without empirical experiments.**
> > > >
> > > >
> > > > **Ans:-**
> > > > In the following, we have included the performance using 2-level iterative teacher, denoted as TGCL$^2$-DSLA (w/ D-SLA). We can see that the performance of TGCL$^2$-DSLA (w/ D-SLA) tends to saturate and often performs worse than TGCL-DSLA (w/ D-SLA).
> > > >
> > > >
> > > > We hypothesize that this observation is primarily because: (a) a better teacher model does not necessarily produce a better student model, and (b) the TGCL-student already recieves the sufficient probability calibrations using the self-supervised DSLA teacher model.
> > > >
> > > >
> > > > |  | BBBP | ClinTox | MUV | HIV | BACE | SIDER | Tox21 | ToxCast | Avg. |
> > > > |---|---|---|---|---|---|---|---|---|---|
> > > > | DSLA | 72.6 $\pm$ 0.8 | 80.2 $\pm$ 1.5 | 76.6 $\pm$ 0.9 | 78.6 $\pm$ 0.4 | 83.8 $\pm$ 1.0 | 60.2 $\pm$ 1.1 | 76.8 $\pm$ 0.5 | 64.2 $\pm$ 0.5 | 74.13 |
> > > > | TGCL-DSLA (w/ D-SLA) | 73.5 $\pm$ 0.9 | **84.9 $\pm$ 0.9** | **79.4 $\pm$ 0.9** | **78.8 $\pm$ 0.5** | **85.2 $\pm$ 0.4** | 61.2 $\pm$ 1.0 | **76.9 $\pm$ 0.1** | **64.9 $\pm$ 0.2** | **75.60** |
> > > > | TGCL$^2$-DSLA (w/ D-SLA) | **74.15$\pm$0.9** | 84.65$\pm$2.0 | 76.43$\pm$1.3 | 78.67$\pm$0.4 | 84.07 $\pm$ 0.8 | **62.73$\pm$ 0.8** | 76.15 $\pm$ 0.4 | 64.32 $\pm$ 0.4 | 75.15 |

---

> > > > > ### Comment · Reviewer_2WDw · 2023-11-23
> > > > > **Response to Ans to Response 2-2,3**
> > > > >
> > > > > I appreciate the experiments, but I'm still unclear about arguments (a) and (b). The claim in phase (a), "a better teacher model does not necessarily produce a better student model," needs more clarity. It's crucial to define what **"better"** means here. Does it refer to improved classification accuracy or closeness to a Bayes teacher?
> > > > >
> > > > > From my understanding, I am convinced that an accurate teacher model doesn't always yield a better student, as accuracy can lead to overconfident, distorted probabilities. In contrast, a model resembling a Bayes teacher could reduce student model variance and enhance performance.
> > > > >
> > > > > In argument (b), the authors suggest the TGCL-student receives sufficient calibration from the DSLA teacher model. If TGCL is close to a Bayes teacher, the theory in [1] suggests that it should ensure strong distillation performance. However, this contradicts the empirical results of TGCL$^2$-DSLA, where it underperforms compared to TGCL.
> > > > >
> > > > > In summary, I can hardly raise my score, since (a) more theoretical analysis (explicit conclusion instead of quoting separated existing works) are needed to illustrate 'when a good student can be distilled from a Bayes-like teacher' (b) how to quantitatively evaluate the distance between the Bayes teacher and the true teacher. (c) How is this method tailored to the graph domain? Since it seems like a standard KD + CL framework that has been witnessed in NLP / CV domains, the paper should emphasize more on what technique is used on the graph domain. (d) the logical derivation in this paper can be further strengthen.

---

> ### Author Response · Authors · 2023-11-23
> **Authros' response**
>
> We appreciate the Reviewer’s response to our rebuttal. We further want to clarify the doubts in the following:
>
> - **It's crucial to define what "better" means here.**
>
> **Ans.** For self-supervised representation learning models, we can define the “better” representation learning model based on their performance in the downstream tasks. However, using such a better representation learning model as a teacher may not lead to a better student model (as suggested in the previous distillation literature).
>
> - **If TGCL is close to a Bayes teacher, the theory in [1] suggests that it should ensure strong distillation performance.**
>
> **Ans.** Please note that TGCL produces better performance. However, it does not necessarily mean that it can be treated as closer to the Bayes teacher.
>
> Further, even when we keep improving the teacher model (i.e., moving closer to the Bayes teacher), we shall reach a saturation point from which we cannot expect any further improvement. Our experimental results suggest that we reach that saturation point after just 1-level of teacher. Otherwise, by iteratively using teacher models, one would be able to achieve 100\% accuracy for any dataset!
>
> -  **more theoretical analysis**
>
> **Ans.** All the required theoretical analyses questioned by the reviewers are already well-known in the literature, making our paper theoretically well-supported. Please refer to our previous responses and the theories cited in our paper.
>
> - **how to quantitatively evaluate the distance between the Bayes teacher and the true teacher.**
>
> **Ans.** Please refer to  A Statistical Perspective on Distillation (ICML 2021) which theoretically analyzed the above-mentioned concern.
>
> - **How is this method tailored to the graph domain?**
>
> **Ans.**
> Please refer to the Introduction (2nd paragraph) and Figure 1 which clearly describe the importance of our proposed method for graph-specific datasets. We briefly summarize these points as follows:
>
> (1) We introduce distilled perceptual distance for graphs where minor changes may lead to major semantical differences. Such a notion of distance is not present in the graph literature.
>
> (2) While existing methods (e.g., DSLA) used the Edit distance, they cannot capture the semantic difference between two graphs with minor changes due to their discrete nature.
>
> (3) Finally, to the best of our knowledge, such distilled perceptual distance is not required for CV or NLP datasets as we can easily apply minor perturbations without changing the semantics drastically.